# Inappropriate Heart Rate Response to Hypotension in Critically Ill COVID-19-Associated Acute Kidney Injury

**DOI:** 10.3390/jcm10061317

**Published:** 2021-03-23

**Authors:** Charles Verney, David Legouis, Guillaume Voiriot, Muriel Fartoukh, Vincent Labbé

**Affiliations:** 1Service de Médecine Intensive Réanimation, Hôpital Tenon, Département Médico-Universitaire APPROCHES, Assistance Publique-Hôpitaux de Paris (AP-HP), 75020 Paris, France; charlesverney@gmail.com (C.V.); guillaume.voiriot@aphp.fr (G.V.); muriel.fartoukh@aphp.fr (M.F.); 2Service de Maladie Infectieuses et Tropicales, Hôpital Tenon, Assistance Publique-Hôpitaux de Paris (AP-HP), 75020 Paris, France; 3Division of Intensive Care, Department of Acute Medicine, University Hospital of Geneva, 1205 Geneva, Switzerland; David.Legouis@unige.ch; 4Laboratory of Nephrology, Department of Medicine, University Hospitals of Geneva, 1205 Geneva, Switzerland; 5Department of Cell Physiology, Faculty of Medicine, University of Geneva, 1205 Geneva, Switzerland; 6Sorbonne Université, 75006 Paris, France; 7Groupe de Recherche Clinique CARMAS, Université Paris Est Créteil, 94000 Créteil, France

**Keywords:** COVID-19, baroreflex, dysautonomic response, critically ill patients

## Abstract

Angiotensin-converting enzyme 2 (ACE2) receptor of severe acute respiratory syndrome coronavirus 2 is involved in baroreflex control mechanisms. We hypothesize that severe coronavirus infectious disease 2019 (COVID-19) patients may show an alteration in baroreflex-mediated heart rate changes in response to arterial hypotension. A pilot study was conducted to assess the response to hypotension in relation to continuous venovenous hemodiafiltration (CVVHDF) in critically ill patients with PCR-confirmed COVID-19 (from February to April 2020) and in critically ill non-COVID-19 patients with sepsis (from February 2018 to February 2020). The endpoint was a change in the heart rate in response to CVVHDF-induced hypotension. The association between COVID-19 status and heart rate change was estimated using linear regression. The study population included 6 COVID-19 patients (67% men; age 58 (53–64) years) and 12 critically ill non-COVID-19 patients (58% men; age 67 (51–71) years). Baseline characteristics, laboratory findings, hemodynamic parameters, and management before CVVHDF-induced hypotension were similar between the two groups, with the exception of a higher positive end-expiratory pressure and doses of propofol and midazolam administered in COVID-19 patients. Changes in the heart rate were significantly lower in COVID-19 patients as compared to critically ill non-COVID-19 patients (−7 (−9; −2) vs. 2 (2;5) bpm, *p* = 0.003), while the decrease in mean arterial blood pressure was similar between groups. The COVID-19 status was independently associated with a lower change in the heart rate (−11 (−20; −2) bpm; *p* = 0.03). Our findings suggest an inappropriate heart rate response to hypotension in severe COVID-19 patients compared to critically ill non-COVID-19 patients.

## 1. Introduction

The binding of severe acute respiratory syndrome coronavirus 2 (o the angiotensin-converting enzyme 2 (ACE2) receptor of host cells leads to ACE2 intracellular pathway downregulation [1]. ACE2 is involved in baroreflex control mechanisms, thus participating in short-term regulation of arterial blood pressure [2]. Therefore, SARS-CoV-2 may alter changes in the heart rate mediated by the baroreflex in response to arterial hypotension in the most severe patients.

Severe acute kidney injury (AKI) is a frequent condition in critically ill coronavirus infectious disease 2019 (COVID-19) patients [3]. AKI participates in a specific phenotype of severe disease, associated with higher hospital mortality and length of stay [3,4]. Acute arterial hypotension is a common side effect of continuous venovenous hemodiafiltration with net ultrafiltration (CVVHDF-induced hypotension [5]) usually used for COVID-19-associated severe AKI [4,6,7].

To investigate potential baroreflex dysfunction in this context, we studied changes in the heart rate in response to CVVHDF-induced hypotension in critically ill COVID-19 patients compared to those in critically ill non-COVID-19 patients.

## 2. Materials and Methods

Two populations of patients requiring CVVHDF in the intensive care unit of Tenon Hospital (Paris, France) were analyzed: patients with polymerase chain reaction-confirmed COVID-19 hospitalized between February and April 2020 and non-COVID-19 patients with sepsis hospitalized from February 2018 to February 2020. Each patient requiring CVVHDF underwent clinical monitoring, including serial hemodynamic parameters (every 2 h) and potential hypotensive episodes. All these data were entered prospectively into a browser-based electronic medical record system and were retrospectively analyzed.

Patients who experienced CVVHDF-induced hypotension during intensive care unit stay were eligible for inclusion. Hypotension was defined as a systolic blood pressure of <90 mmHg with a >20 mmHg decrease (or a mean blood pressure of <65 mmHg with a >10 mmHg decrease) [5]. The reduction in blood pressure was equal to the blood pressure at the time of CVVHDF-induced hypotension minus the blood pressure at the time of the last hemodynamic monitoring before CVVHDF-induced hypotension (maximum of 2 h before). Exclusion criteria were non-sinus rhythm, drug-induced bradycardia upon onset of CVVHDF-induced hypotension (cardiovascular medication, including betablockers, neuroleptic, dexmedetomidine), neurologic disease, and any therapeutic modification among ventilation modes or settings, fluid loading, ultrafiltration, vasopressors, and sedative drugs within the 2 h preceding CVVHDF-induced hypotension. Baseline clinical characteristics, clinical and laboratory findings, management, and hemodynamic parameters before the onset of CVVHDF-induced hypotension were collected. The endpoint was the change in the heart rate defined as the heart rate at the time of CVVHDF-induced hypotension onset minus the heart rate at the time of the last hemodynamic monitoring before CVVHDF-induced hypotension (maximum of 2 h before).

Variables for individuals were expressed as the median and 25th–75th percentile or absolute and relative (%) frequencies and compared using the Wilcoxon rank sum test or the Fisher exact test, where appropriate. The association between COVID-19 status and heart rate changes was estimated using linear regression. To take into account confounding factors, multivariable analysis was performed using linear regression with heart rate change as a dependent variable. For the selection of independent variables, two strategies were used. In the first model, we included significantly different variables in univariable analysis if clinically relevant, fitting a parsimonious model in line with the small sample size. In the second model, we first included every variable with a univariable *p*-value less than 0.2 and ran a stepwise backward selection. Interactions between variables were assessed in both models, and assumptions were checked. *p*-Values of less than 0.05 were considered significant, and all *p*-values were two tailed. Statistical analysis was performed using R software.

The study was approved by the ethical board of the French Intensive Care Society (CE SRLF, 20–80).

## 3. Results

Overall, 6 COVID-19 patients (67% men; age 58 (53–64) years) and 12 critically ill non-COVID-19 patients with sepsis (58% men; age 67 (51–71) years) were included (Figure 1).

Baseline characteristics, clinical and laboratory findings, and hemodynamic parameters, before CVVHDF-induced hypotension were similar between the two groups. Regarding the management before CVVHDF-induced hypotension, positive end-expiratory pressure (PEEP) was higher in COVID-19 patients (10.0 (8.0–10.5) cmH_2_0 vs. 6.0 (5.3–6.0) cmH_2_0, *p* = 0.002), as were propofol and midazolam dosages (200 (200–200) mg/h vs. 160 (75–200) mg/h, *p* = 0.02 and 4.0 (0.8–5.0) mg/h vs. 0 (0–0) mg/h, *p* = 0.02, respectively) (Table 1).

The time of occurrence of CVVHDF-induced hypotension from CVVHDF initiation was similar between COVID-19 and non-COVID-19 patients (4 days (1–7) vs. 5 days (2–7), *p* = 0.70). Heart rate change was significantly lower in COVID-19 patients as compared to non-COVID-19 patients (−7 (−9–−2) vs. 2 (2–5) bpm, *p* = 0.003; Figure 2a), while the decrease in mean arterial blood pressure was similar between the two groups (Figure 2b). Intercepts between mean arterial blood pressure change and heart rate change fitted by linear regression were different between COVID-19 and non-COVID-19 patients (*p* = 0.004; Figure 2c).

After adjustment of PEEP, and dosage of propofol and midazolam using multivariable linear regression, the COVID-19 status was independently associated with a lower change in the heart rate (−11 (−20–−2) bpm; *p* = 0.03; Table 2). To perform sensitivity analysis, we carried out a second linear regression using a stepwise backward selection. The COVID-19 status remained independently associated with a lower change in the heart rate (−12 (−20–−5) bpm, *p* = 0.003; Appendix A).

## 4. Discussion

In this pilot study, COVID-19 was associated with an inappropriate decrease in the heart rate in response to CVVHDF-induced hypotension in critically ill patients. While numerous COVID-19-related cardiovascular alterations such as myocarditis, arrhythmias, and bradycardia have already been reported in critically ill patients, an inappropriate heart rate response to hypotension has not yet been described [8].

As previously reported in many other viral infections, such as human immunodeficiency virus and herpes viruses [9], involvement of the autonomic nervous system with baroreflex dysfunction could explain this phenomenon. The heart rate response is predominantly regulated through the baroreflex control of the balance between sympathetic and parasympathetic inputs. In healthy subjects, the increased compensatory tachycardia in response to arterial hypotension induces higher cardiac output, while the stroke volume remains unchanged. A dysfunction of this autonomic adaptive mechanism in critically ill patients with COVID-19-associated severe AKI could lead to inadequate organ perfusion, thus participating in the development of multi-organ failure. Similarly, attenuation of autonomic function in patients with sepsis and multiple-organ dysfunction syndrome is associated with poor outcomes [10]. Therefore, increased awareness of recognizing the risk of inadequate cardiovascular regulation in those severe COVID-19 patients is needed. Nonetheless, further studies are needed to confirm dysautonomia in severe COVID-19 patients as well as to investigate its clinical and prognostic impact.

The underlying mechanisms of potential baroreflex dysfunction in patients with COVID-19-associated severe AKI are unknown. The baroreceptor reflex control of the heart rate is regulated by the brain renin–angiotensin system, located mainly in the nucleus of the solitary tract of the brainstem, where ACE2 is expressed [2,11]. SARS-CoV-2 may cause ACE2 internalization and downregulation in the nucleus of the solitary tract, thus altering the baroreflex response and partly inhibiting the compensatory increase in the heart rate during acute arterial hypotension. Some preclinical evidence supports this hypothesis. Studies have shown that inhibition of ACE2 activity in the nucleus of the solitary tract, through injection of a specific inhibitor in this region, reduces the sensitivity of the baroreceptor reflex control of the heart rate [12]. Similarly, Xia et al. described impaired baroreflex and autonomic function in ACE2-knockout mice [13]. However, in the non-COVID-19 context, AKI has already been shown to associate with autonomic nervous dysfunction. Levitan et al. showed that patients with AKI have reduced arterial blood pressure during orthostatic stress without an appropriate rise in the heart rate [14]. To what extent the potential baroreflex dysfunction in COVID-19 patients is related to SARS-CoV-2 infection alone or combined with AKI-related pathways requires further investigation.

Our study had several limitations. First, this was a single-center investigation with a limited number of patients. Second, only patients undergoing net ultrafiltration during CVVHDF, thus reflecting a relative hemodynamic stability before acute arterial hypotension, were eligible. Third, our results need to be confirmed in COVID-19 patients not undergoing renal replacement therapy.

## 5. Conclusions

We report an inappropriate heart rate response to CVVHDF-induced hypotension in intensive care COVID-19 patients, probably due to baroreflex dysfunction. Therefore, increased awareness regarding the role of this cardiovascular regulatory mechanism is needed when managing these patients.

## Figures and Tables

**Figure 1 jcm-10-01317-f001:**
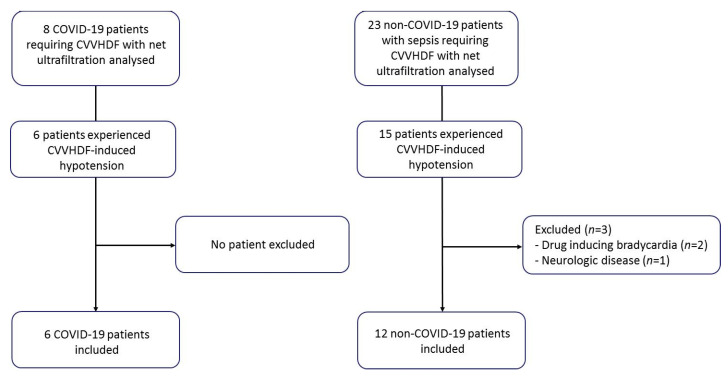
Flowchart of the study. Abbreviations: COVID-19, coronavirus infectious disease 2019; CVVHDF, continuous venovenous hemodiafiltration.

**Figure 2 jcm-10-01317-f002:**
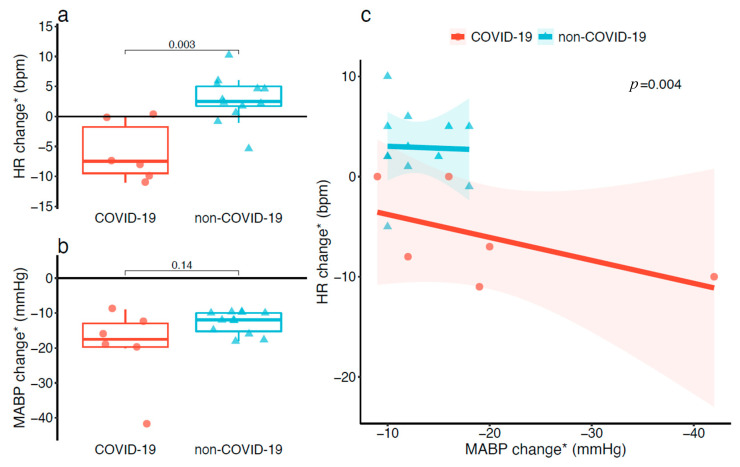
Changes in the heart rate (**a**) and mean arterial blood pressure (**b**) during CVVHDF-induced hypotension, contrasting COVID-19 patients and non-COVID-19 patients. Linear regression (**c**) showing a non-physiological association between mean arterial blood pressure change and heart rate change. Abbreviations: COVID-19, coronavirus infectious disease 2019; CVVHDF, continuous venovenous hemodiafiltration; HR, heart rate; MABP, mean arterial blood pressure. * Parameters at the time of CVVHDF-induced hypotension onset minus parameters at the time of last hemodynamic monitoring before CVVHDF-induced hypotension (maximum of two hours before).

**Table 1 jcm-10-01317-t001:** Patients’ baseline characteristics, clinical and laboratory findings, management before CVVHDF-induced hypotension, and description of CVVHDF-induced hypotension between COVID-19 patients and non-COVID-19 patients.

Variables	All Patients (*n* = 18)	COVID-19 Patients (*n* = 6)	Non-COVID-19 Patients (*n* = 12)	*p*-Value
Baseline clinical characteristics				
Gender (male)	12 (66.7)	5 (83.3)	7 (58.3)	0.60
Age, years	63 (53–69)	58 (53–64)	67.0 (51–71)	0.30
Arterial hypertension	12 (66.7)	5 (83.3)	7 (58.3)	0.60
Diabetes mellitus	7 (38.9)	3 (50.0)	4 (33.3)	0.63
Congestive heart failure	1 (5.6)	0	1 (8.3)	0.47
Chronic dialysis	3 (16.7)	2 (33.3)	1 (8.3)	0.24
Clinical and laboratory findings *				
Core temperature, °C ^†^	37.7 (36.3–37.0)	36.7 (36.4–36.9)	36.8 (36.3–37.1)	0.96
Richmond Agitation Sedation Scale	−5.0 (−5.0–−0.8)	−5.0 (−5.0–0.0)	−4.5 (−5.0–−1.5)	0.88
Natremia, mmol/L	141 (138–142)	139 (138–140)	141 (140–142)	0.24
Kaliemia, mmol/L	4.0 (3.8–4.2)	3.8 (3.7–4.0)	4.1 (3.9–4.2)	0.16
Calcemia, mmol/L	2.46 (2.37–2.61)	2.51 (2.43–2.70)	2.43 (2.36–2.59)	0.48
Magnesemia, mmol/L	1.00 (0.91–1.06)	0.98 (0.91–1.01)	1.01 (0.93–1.08)	0.51
Phosphatemia, mmol/L	1.43 (1.32–1.60)	1.46 (1.31–1.68)	1.43 (1.37–1.54)	0.85
Management ^†^				
Invasive mechanical ventilation	18 (100)	6 (100)	12 (100)	>0.99
Volume-controlled ventilation	12 (67)	4 (67)	8 (67)	>0.99
Pressure support ventilation	6 (33)	2 (33)	4 (33)	>0.99
PEEP, cmH_2_0	6.0 (6.0–8.5)	10.0 (8.0–10.5)	6.0 (5.3–6.0)	0.002
PaO_2_/FiO_2_	157 (122–225)	140 (112–128)	182 (128–239)	0.42
Tidal volume, mL/kg PBW ^‡^	6.3 (5.8–6.8)	6.4 (5.8–7.0)	6.1 (5.9–6.5)	0.88
Catecholamine				
Norepinephrine	9 (50.0)	3 (50.0)	6 (50.0)	>0.99
Norepinephrine dose, µg/kg/min	0.12 (0.10–0.20)	0.12 (0.11–0.13)	0.16 (0.11–0.31)	0.71
Epinephrine	1 (5.6)	0	1 (8.3)	>0.99
Epinephrine dose, µg/kg/min	0.2	0	0.2	>0.99
CVVHDF parameters				
Ultrafiltration rate, mL/h	100 (100–172)	100 (100–137)	125 (50–250)	0.92
Blood flow rate, mL/min	150 (150–150)	150 (150–153)	150 (150–150)	0.91
Dialysate flow rate, mL/h	3000 (2700–3150)	3000 (3000–3200)	3000 (2875–3050)	0.39
Sedative drugs				
Propofol	17 (94.4)	6 (100)	11 (91.7)	0.92
Propofol dose, mg/h	200 (150–200)	200 (200–200)	160 (75–200)	0.02
Sufentanil	18 (100)	6 (100)	12 (100)	>0.99
Sufentanil dose, µg/h	20 (10.0–20)	20 (20–27)	20 (10–20)	0.18
Midazolam	6 (33.3)	4 (66.7)	2 (16.7)	0.11
Midazolam dose, mg/h	0.0 (0.0–2.8)	4 (0.8–5.0)	0 (0–0)	0.02
Ketamine	1 (5.6)	0 (0.0)	1 (8.3)	>0.99
Ketamine dose, mg/h	0 (0–0)	0 (0–0)	0 (0–0)	0.48
Neuromuscular-blocking agents	9 (50.0)	5 (83.3)	4 (33.3)	0.13
Hemodynamic parameters				
Before CVVHDF-induced hypotension ^†^				
Heart rate, bpm	108 (84–120)	114 (85–119)	100 (89–120)	0.89
MABP, mmHg	74 (70–75)	78 (73–79)	72 (67–74)	0.08
SABP, mmHg	118 (105–127	122 (117–128)	117 (105–124)	0.40
DABP, mmHg	54 (51–57)	57 (53–60)	53 (50–54)	0.24
At the time of CVVHDF-induced hypotension onset				
Heart rate, bpm	106 (92–117)	106 (83–110)	104 (91–120)	0.45
MABP, mmHg	58 (56–60)	58 (57–60)	58 (56–62)	0.67
SABP, mmHg	89 (84–97)	88 (81–99)	91 (86–96)	>0.99
DABP, mmHg	46 (41–48)	45 (42–48)	45 (41–47)	0.67
Changes (Δ) in ^§^				
Heart rate, bpm	1.5 (−4–4.5)	−7 (−9–−2)	2.5 (2–5)	*p* = 0.003
MABP, mmHg	−13.5 (−18–−10.5)	−18 (−20–−16)	−12 (−15–−10)	*p* = 0.12
SABP, mmHg	−24.5 (−32–−21)	−24.5 (−35.5–−23)	−24.5 (−29–−20)	*p* = 0.42
DABP, mmHg	−9 (−12–−5)	−10.5 (−13–−6)	−8 (−10.5–−5.5)	*p* = 0.57

Continuous variables are expressed as medians (25th–75th percentile). Categorical variables are expressed as numbers (percentages). Definitions of abbreviations: COVID-19, coronavirus infectious disease 2019; CVVHDF, continuous venovenous hemodiafiltration; DABP, diastolic blood pressure; FiO2, fraction of inspired oxygen; PaO2, partial pressure of oxygen; MABP, mean arterial blood pressure; PEEP, positive end-expiratory pressure; PBW, predicted body weight; RASS, Richmond Agitation Sedation Scale; SABP, systolic arterial blood pressure. * On the day of CVVHDF-induced hypotension onset. ^†^ At the time of last hemodynamic monitoring before CVVHDF-induced hypotension (maximum of two hours before). ^‡^ In patients requiring volume-controlled ventilation. ^§^ Parameters at the time of CVVHDF-induced hypotension onset minus parameters at the time of last hemodynamic monitoring before CVVHDF-induced hypotension (maximum of two hours before).

**Table 2 jcm-10-01317-t002:** Linear regression with parsimonious strategy assessing the influence of the COVID-19 status and covariates on the change in the heart rate during CVVHDF-induced hypotension in critically ill patients.

Variables	Estimated * Heart Rate Change ^†^, bpm	95% CI	*p*-Value
COVID-19 status	−10.7	−19.8–−1.6	0.03
Midazolam dose ^‡^	−0.5	−1.5–0.5	0.30
Propofol dose ^‡^	0.0	−0.0–0.1	0.66
PEEP ^‡^	0.9	−1.2–2.9	0.38

Definitions of abbreviations: CI, confidence interval; COVID-19, coronavirus infectious disease 2019; CVVHDF, continuous venovenous hemodiafiltration; PEEP, positive end-expiratory pressure. * Estimate-adjusted COVID-19 status and covariates (PEEP, propofol dose, and midazolam dose). Estimates represent the difference in the heart rate changes in COVID-19 patients as compared to non-COVID-19 patients. ^†^ Heart rate at the time of CVVHDF-induced hypotension onset minus heart rate at the time of last hemodynamic monitoring before CVVHDF-induced hypotension (maximum of two hours before). ^‡^ At the time of last hemodynamic monitoring before CVVHDF-induced hypotension (maximum of two hours before).

## Data Availability

All data and materials fully comply with field standards and might be available upon reasonable request.

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
