# Peer review of "Inappropriate Heart Rate Response to Hypotension in Critically Ill COVID-19-Associated Acute Kidney Injury"

_jcm, 2021, doi:10.3390/jcm10061317_

Round 1
Reviewer 1 Report
I read your manuscript on heart rate response in critically ill COVID-19 patients with great interest. While your findings seem interesting for intensive care unit clinicians, some points need to be clarified in order to further evaluate your study.
-) Overall manuscript: English style and grammar editing are necessary throughout the whole manuscript. Often, articles are missing, or sentences sound strange – I suggest editing by a native speaker.
-) Abstract: first sentence: Is this known from literature or do you hypothesize this? I known you describe this in your Introduction section, but the statement is somewhat unclear in the abstract.
-) Methods: You define CVVHDF-induced hypotension as a blood pressure under a certain value and a fall of certain mmHg points; however, you do not state in which timeframe these changes could occur in order to become eligible for inclusion?
-) Methods: I suppose you retrospectively collected data from your clinical patient forms and electronic records? You do not mention this, please provide much more details about your actual methods of data acquisition and processing.
-) Methods: You list a few exclusion criteria, for instance a change in vasopressors; however, what about changes in ventilation modes or settings?
-) Methods: Why did you only include so few patients, when your ICU surely treats more critically ill patients requiring CVVHDF? Could there be any selection bias?
-) Methods: A detailed description of your statistical methods is missing.
-) Table 1: Please provide more details on your ventilation modes and -settings. Since you describe so few patients, these details should be easy to obtain, and would complete the basic characteristics. Also, one could identify any ventilation-associated factors influencing your results.
-) Results: Please provide information on the patients’ sedation levels (all RASS -5? If not, are they comparable in terms of heart rate as stress is difficult to objectify?)
-) Results: Why was no multivariate regression analysis conducted? How would you estimate the potential biasing influence of all listed factors and parameters on your primary outcome?
-) Your Discussion section is mainly repeating your Introduction and does not give any new information which would be interesting for a clinician. Please expand your Discussion section appropriately.
-) Conclusion: Please also state that the mentioned hypotensive episode derived from CVVHDF.
Author Response
Response to Reviewer 1 Comments
Reviewer 1
C1R1 Overall manuscript: English style and grammar editing are necessary throughout the whole manuscript. Often, articles are missing, or sentences sound strange – I suggest editing by a native speaker.
Response Thank you for your valuable comment. We proofread our manuscript by an English native speaker
C2R1 Abstract: first sentence: Is this known from literature or do you hypothesize this? I known you describe this in your Introduction section, but the statement is somewhat unclear in the abstract.
Response We agree with your opinion. We suggest to amend this first sentence with the following one: “Angiotensin-converting enzyme 2 (ACE2) receptor of SARS-CoV-2 is involved in baroreflex control mechanisms. We hypothesize that severe COVID-19 patients may have an alteration in baroreflex mediated heart rate changes in response to arterial hypotension”
C3R1 Methods: You define CVVHDF-induced hypotension as a blood pressure under a certain value and a fall of certain mmHg points; however, you do not state in which timeframe these changes could occur in order to become eligible for inclusion?
Response Thank you for your valuable comment. Patients who experienced CVVHDF-induced hypotension during ICU stay were eligible for inclusion regardless of the time from the initiation of CVVHDF. We clarified this point by adding the following sentence in the methods section, page 3, lines 8-9: “Patients who experienced CVVHDF-induced hypotension during ICU stay were eligible for inclusion”
In addition, we defined the reduction in blood pressure as following in the methods section, page 3 lines 11-13: “The reduction in blood pressure was equal to the blood pressure at the time of CVVHDF-induced hypotension minus the blood pressure at the time of the last hemo-dynamic monitoring before CVVHDF-induced hypotension (maximum two hours before)”.
C4R1 Methods: I suppose you retrospectively collected data from your clinical patient forms and electronic records? You do not mention this, please provide much more details about your actual methods of data acquisition and processing.
Response Thank you for your valuable comment. We clarified this point by changing the following sentence in the methods section: “Serial hemodynamic parameters (every 2-hours) and potential episodes of hypotension reported on medical records were retrospectively analysed » by this one, in the methods section, page 2 lines 11-14 “Each patient requiring CVVHDF underwent clinical monitoring, including serial hemodynamic parameters (every 2 hours) and potential hypotensive episodes. All these data were entered prospectively into a browser-based electronic medical record system and were retrospectively analyzed.”
C5R1Methods: You list a few exclusion criteria, for instance a change in vasopressors; however, what about changes in ventilation modes or settings?
Response We thank the referee for her/his comment. We agree that ventilator settings impact hemodynamics patient. We apologize for this omission in the list of therapeutic modification before the CVVHDF induced hypotension defining the exclusion criteria. We clarified this point in the methods section page 2 lines 21-25: “Exclusion criteria were non-sinus rhythm, drug-induced bradycardia upon onset of CVVHDF-induced hypotension (cardio-vascular medication including betablockers, neuroleptic, dexmedetomidine), neurologic disease, and any therapeutic modification among ventilation modes or settings, fluid loading, ultrafiltration, vasopressors, sedative drugs within the 2 hours preceding CVVHDF-induced hypotension”. Thus, none of the included patients had undergone a change in ventilation modes or settings before the CVVHDF-induced hypotension.
C6R1: Methods: Why did you only include so few patients, when your ICU surely treats more critically ill patients requiring CVVHDF? Could there be any selection bias?
Response We agree with your opinion. Firstly, intermittent hemodialysis is frequently prefedred over CVVVHDF in our ICU. Secondly, among patient requiring CVVHDF, patients without net ultrafiltration were not eligible for inclusion as it could reflect a precarious hemodynamic status and thus introduce biases in our analyses. We propose to add this point in the limits of the study, page 6 lines 36-42. Finally we clarify this point by adding “with net ultrafiltration” in the introduction section page 1 line 45, and the flow chart of the study. We can add on the top of the flow chart the number of patients (COVID-19 and non-COVID-19) who required CVVHDF regardless of the net ultrafiltration administration if the examiner wishes.
C7R1 Methods: A detailed description of your statistical methods is missing.
Response We apologize for this omission. We added this following paragraph in the method section page 2 lines 31-43. “Variables for individuals were expressed as median and 25th–75th percentile or ab-solute and relative (%) frequencies and compared using Wilcoxon rank sum test or Fisher exact test, where appropriate. The association between COVID-19 status and heart rate changes was estimated using linear regression. In order to take into account confounding factors, a multivariable analysis was performed using linear regression with heart rate change as a dependent variable. For the selection of independent variables, two strategies were used. In the first model, we included significantly different variables in univariable analysis if clinically relevant, fitting a parsimonious model in line with the small sample size. In the second model, we first included every variable with a univariable p value less than 0.2 and ran a stepwise backward selection. Interactions between variables were as-sessed in both models and assumptions were checked. P values of less than 0.05 were considered as significant, and all p values were two tailed. Statistical analysis was per-formed using R software.”
C8R1 Table 1: Please provide more details on your ventilation modes and -settings. Since you describe so few patients, these details should be easy to obtain, and would complete the basic characteristics. Also, one could identify any ventilation-associated factors influencing your results.
Response Thank you for this clever reviewing. We added details about ventilation modes and settings in Table 1. While PEEP was significantly higher in COVID-19 patients, comparing to non-COVID-19 patients the ventilator mode and the PaO2/FIO2 ratio were did not differ. We added this result in the results section page 3lines 5-8 with the following statement:” Regarding the management before CVVHDF-induced hypotension, positive end-expiratory pressure (PEEP) was higher in COVID-19 patients (10.0 [8.0–10.5] cmH20 vs.6.0 [5.3–6.0] cmH20, p=0.002)”. We thus included PEEP variable in the first multivariable linear regression: After adjustment on PEEP and dosage of propofol and midazolam, COVID-19 status was independently associated with a lower heart rate change. We have changed the original model to this new model in Table 2. In addition, as sensitivity analysis, we carried out a second linear regression using a stepwise backward selection. COVID-19 status remained independently associated with a lower change in heart rate (Table S1).
C9R1 Results: Please provide information on the patients’ sedation levels (all RASS -5? If not, are they comparable in terms of heart rate as stress is difficult to objectify?)
Response We thank the referee for her/his comment. The RASS before CCVHF induced hypotension was similar among groups (-5.0 [-5.0 – 0.0] vs -4.5 [-5.0 – -1.5] p=0.88). We report these results in Table 1 in the clinical findings section. In the same line, heart rate before CVVHDF-induced hypotension did not differ accross groups (Table 1 in the hemodynamic parameters section)
C10R1 Results: Why was no multivariate regression analysis conducted? How would you estimate the potential biasing influence of all listed factors and parameters on your primary outcome?
Response We thank the referee for her/his comment. We performed a multivariable linear regression to fit the relationship between the heart rate change according to COVID-19 status, adjusted by the following confounding factors: PEEP, midazolam and propofol doses. Those variables were selected as they were significantly different among groups in univariable analyses and clinically relevant. In this model, COVID-19 status was independently associated with a lower heart rate change in response to CVVHDF-Induced Hypotension (Table 2). We added this precision in the result section.
As a sensitivity analyses, we performed in the revised manuscript a second multivariable linear regression, built without a priori hypothesis. Briefly, variables with a univariate p-value < 0.2 were first considered in the multivariable analysis. The final regression model was thus performed using stepwise backward selection. In this model, COVID-19 status was still independently associated with a lower heart rate change in response to CVVHDF-induced hypotension. Other selected variables included MABP change and PEEP. We added this sensitivity analysis as following in Supplementary files of the manuscript (Table S1)
|
Variables |
Estimate † heart rate change, bpm |
95% CI |
p value |
|
COVID-19 status |
- 12.12 |
-19.52 – -4.72 |
0.003 |
|
MABP change * |
- 0.27 |
-0.57 – 0.03 |
0.07 |
|
PEEP ‡ |
1.42 |
-0.38 – 3.22 |
0.11 |
C11R1 Your Discussion section is mainly repeating your Introduction and does not give any new information which would be interesting for a clinician. Please expand your Discussion section appropriately.
Response We agree with your opinion. As asking by reviewer, we expanded the discussion section. In particular, we add the following paragraph in the discussion section with information of interest for a clinician: “As previously reported in many others viral infections, such as HIV and herpes viruses [7], involvement of the autonomic nervous system with baroreflex dysfunction could explain this phenomenon. Heart rate response is predominantly regulated through the baroreflex control of the balance between sympathetic and parasympathetic inputs. In healthy subjects, the increased compensatory tachycardia in response to arterial hypotension induces higher cardiac output, while stroke volume remains unchanged. A dysfunction of this autonomic adaptive mechanism in severe COVID-19 patients could lead to inadequate organ perfusion thus participating in the development of multi-organ failure. Similarly, attenuation of autonomic function in patients with sepsis and multiple organ dysfunction syndrome is associated with poor outcomes [8]. Therefore, increased awareness in recognizing the risk of inadequate cardiovascular regulation in severe COVID-19 patients is needed. Nonetheless, further studies are needed to confirm dysautonomia in severe COVID-19 patients as well as to investigate its clinical and prognostic impact”, page 6, lines 14-27.
C12R1 Conclusion: Please also state that the mentioned hypotensive episode derived from CVVHDF
Response Thank you for your comment. We added this precision page 7, lines 2-5: “We report an inappropriate heart rate response to CVVHDF-induced hypotension in intensive care COVID-19 patients, probably due to baroreflex dysfunction”.

Reviewer 2 Report
The article briefly describes an interesting retrospective observation with inclusion and exclusion criteria for the study and with a control group - hemodynamic disorders consisting in a drop in blood pressure and accompanying slow heart rate after switching on CVVHD in patients with COVID-19. (usually the opposite reaction in patients with a healthy heart, the normal response to a fall in blood pressure and hypovolemia is an increase in heart rate). The difference in heart rate was statistically confirmed in a small 6-person COVID group compared to a similar 12-person non-COVID group. The used anesthetics ketamine and propofol and catecholamines epinephrine (adrenaline) and norepinephrine change the heart rate and the response from baroreceptors (epinephrine was used in the non-covid group) but they were used similarly in both groups and no statistically evident difference between the groups was shown in their use. I believe that this is an interesting observational study of COVID-19 patients using CVVHDF.
A question: why epinephrine was not used in patients with COVID-19 in response to hypotension and bradycardia after starting CVVHDF?
Author Response
Response to Reviewer 2 comments
C1R2 why epinephrine was not used in patients with COVID-19 in response to hypotension and bradycardia after starting CVVHDF?
Response In this study, our aim was to investigate if COVID-19 status was associated to a lower heart rate change in response to dialysis-induced hypotension. For this purpose, we only focused on the variables recorded before and during DIH but not in the management itself of this hypotensionHowever, we can insert the management of the CVVHDF induced hypotension as following in Table 1, depending on the referee preference.
|
Variables |
All patient (n=18) |
COVID-19 patients (n=6) |
Non COVID-19 patients (n=12) |
p value |
|
Decreased net ultrafiltartion rate |
3 (17) |
1 (17) |
(2) 17 |
>0.99 |
|
Stop net ultrafiltration |
11 (61) |
4 (67) |
7 (58) |
0.73 |
|
Intravenous fluid bolus |
3 (17) |
0 |
3 (25) |
0.18 |
|
Initiation or intensified Norepinephrine |
6 (33) |
2 (33) |
4 (33) |
>0.99 |
|
Initiation or intensified Epinephrine |
1 (6) |
0 |
1 (8) |
0.56 |

Reviewer 3 Report
The aim of the present study was to examine an effect of COVID-19 on HR response to CVVHDF-induced hypotension in patients to assess baroreflex function in the patients who used CVVHDF as continuous renal replacement therapy. The main finding that COVID-19 attenuates HR response to CVVHDF-induced hypotension is clear and important information in this patients. However, the research significance of this study is unclear. The authors should clarify the suggestions of this finding in this clinical area for readers. For example, the authors need to provide the clear reason why baroreflex dysfunction is problem in this patients.
Another question is that it is well known that patients with renal failure attenuate baroreflex function. For example, these patients cause orthostatic intolerance. Thus, regardless of having COVID-19, baroreflex function attenuates in these patients, indicating that these patients need to be careful for inadequate cardiovascular regulation. The finding of the present study demonstrated that COVID-19 accelerates baroreflex dysfunction. How is this suggestion important in this clinical area?
The phenomena that COVID-19 attenuates cardiac baroreflex function via the modified ACE2 is interesting. This study need to confirm it in healthy subjects rather than patients but it is unclear why the authors test it in this patients for the above reasons.
Author Response
Response to reviewer 3 comments
C1R3 the research significance of this study is unclear. The authors should clarify the suggestions of this finding in this clinical area for readers. For example, the authors need to provide the clear reason why baroreflex dysfunction is problem in these patients.
C2R3 Another question is that it is well known that patients with renal failure attenuate baroreflex function. For example, these patients cause orthostatic intolerance. Thus, regardless of having COVID-19, baroreflex function attenuates in these patients, indicating that these patients need to be careful for inadequate cardiovascular regulation. The finding of the present study demonstrated that COVID-19 accelerates baroreflex dysfunction. How is this suggestion important in this clinical area?
Response (to C1R3 and C2R3)
We thank reviewer 3 for this comment. As asked by reviewer, we added the following paragraph in the discussion section: “As previously reported with many others viral infections, such as HIV and herpes viruses [7], involvement of autonomic nervous system with baroreflex dysfunction could explain this phenomenon. Indeed, heart rate response is predominantly regulated through the baroreflex control of the balance between sympathetic and parasympathetic inputs. In healthy subjects, the increased compensatory tachycardia in response to arterial hypotension induces higher cardiac output, while stroke volume remains unchanged. A dysfunction of this autonomic adaptive mechanism in severe COVID-19 patient could lead to an inadequate organs perfusion in response to arterial variability and finally participate to the multi-organ failure. In the same way, autonomic function of patients with sepsis and multiple organ dysfunction syndrome is blunted, and this attenuation is associated with poor outcomes [8]. So intensivists need to be careful for the risk of inadequate cardiovascular regulation and of multi-organ failure in severe COVID-19 patients. Whatever, further studies are required to confirm dysautonomia in severe COVID-19 and investigate its clinical and prognosis impact, page 6, lines 14-27.
C3R3 The phenomena that COVID-19 attenuates cardiac baroreflex function via the modified ACE2 is interesting. This study needs to confirm it in healthy subjects rather than patients but it is unclear why the authors test it in this patients for the above reasons.
Response
We thank the referee for her/his comment and agree with her.him. We chose this specific population to investigate potential baroreflex dysfunction because (i) the need for renal replacement therapy in critically ill COVID-19 patient may reflect a particular phenotype of severe infection with poor prognosis and (ii) acute arterial hypotension is common in this setting. We clarified this point by adding the following sentence in the rational of the study: “Moreover the need for renal replacement therapy in critically ill COVID-19 patients may reflect a particular phenotype of severe infection with poor prognosis [5]”, page 1, lines 1-3. In addition, we added the following sentence in the limit of the study “Third, our results need to be confirmed in COVID-19 patients not undergoing renal replacement therapy”, page 6, lines 41-42.

Round 2
Reviewer 3 Report
Thank you for your work. I do not have further comments.
Author Response
Sir,
We would like to thank you for your valuable time and input.
Sincerely Yours
Vincent Labbé, MD